

# Calibrating Comfort in the Care-Control nexus of Humanitarian Borderwork

Bronte Alexander[1]

**1** Griffith University
b.alexander@griffith.edu.au

Wednesday, February 5, 25

## Abstract

People seeking safety through migration often move through spaces of humanitarian care, including shelters, camps, and transit sites. Despite differences in resources and infrastructure, issues with overcrowding and poor thermal conditions persist globally. These concerns are directly linked to the degree in which comfort is considered in the design of humanitarian and emergency spaces. Drawing on research regarding Brazil's military-humanitarian response to Venezuelan migration, this paper explores the often-overlooked role of comfort in humanitarian borderwork. By studying the operations and spatialities of two sites that provide care to incoming migrants and refugees, I argue that comfort can be used as a lens to examine the function of care and control in humanitarian borderwork. In doing so, this research highlights how (dis)comfort works through both care and control to (re)produce differential and restrictive mobilities. Challenging the normalisation and invisibilisation of discomfort in displacement contexts reveals the need to further consider the everyday, mundane implications of care.

## 1. Introduction

In emergency planning, comfort is rarely taken into consideration. Providing access to basic shelter, food, and healthcare take precedent. Forced or internal displacement requires multi-scalar coordination regardless of whether increased (im)mobility is prompted by climate, conflict, or economic collapse. It is indeed an enormous task. As a result, emergency shelter is typically designed as a temporary solution (Aburamadan, 2022). It requires straightforward construction and order, ensuring clear oversight and management of the space (Martin et al., 2020). However, shelters and camps across the globe continue to face challenges related to overcrowding, limited hygiene facilities, and poor thermal conditions and air quality (Conzatti et al., 2022; Elorduy, 2017). By focusing on managing mobilities efficiently (Pallister-Wilkins, 2018a) and perpetuating the least of all possible evils (Weizman, 2011), providing comfort – even in the most basic sense – is often neglected.

This research interrogates the role of comfort in humanitarian borderwork. By challenging the invisiblisation and normalisation of discomfort in displacement contexts, I argue that comfort emerges as a possible lens through which we can examine the care-control nexus of humanitarianism. Using comfort as an analytical lens allows us to explore the inherent exclusion underscoring humanitarian borderwork. I show how (dis)comfort works through care and control to (re)produce both differential and restrictive mobilities. While comfort can often be understood as a sense or experience, here, I focus on the spatialities of comfort and its role in mobility governance.

To tackle the question of comfort, I investigate Brazil's recent and ongoing response to Venezuelan forced migration – what is considered by many international leaders to be an exemplar of humanitarianism (UN News, 2019). Their efforts have been somewhat celebrated, and not without reason. The federal humanitarian task force 'Operação Acolhida' or 'Operation Welcome' has been instrumental in providing support to vulnerable groups, building structures for humanitarian spaces and shelters, distributing food and goods, and issuing Taxpayer Identification Number (CPF) cards that enable Venezuelans to work in Brazil (Presidência da República: Casa Civil, n.d.). Despite these accolades, the challenges of delivering care to considerably 'at risk' or 'vulnerable' populations remain. Particularly when those same groups are simultaneously framed as potential risks to the 'host' society – much like other migrants, refugees, and people seeking asylum around the world.

Due to economic and political instability, Venezuelans have been leaving the country in record numbers. By late 2023, there were over 7.7 million Venezuelan migrants and refugees worldwide, with the majority remaining in the Latin American and Caribbean region (UNHCR, n.d.). Brazil saw steady increases in Venezuelan migration from 2015, and by the time of this research in 2019, there were an estimated 500 people crossing the shared land border each day (Zapata & Tapia Wenderoth, 2021). During this period, there were over 280,000 Venezuelans in Brazil (Response for Venezuelans, 2019), many of whom were granted entry as refugees on a *prima facie* basis under a broader Cartagena Declaration definition (Ochoa, 2020, p. 489). Those who arrived at the border in Brazil were likely to have limited sources of income in Venezuela and lower educational attainment than those able to travel to further distant countries (Casey, 2016; IOM, 2020). In response to the growing number of Venezuelan arrivals, the Operation Welcome task force – supported by several government departments, UN agencies, (inter)national organisations, and civil society groups – works to provide food, shelter, border security, and internal settlement services (Response for Venezuelans, 2019).

The scale of the Operation Welcome response is impressive. By mid-2019 the Brazilian government had spent R$265 million (about USD$46 million) on military and humanitarian intervention in the northern state of Roraima, which borders Venezuela (Jovem Pan, 2019). This injection of funds can be visibly seen in the border city of Pacaraima and the capital of Boa Vista, where a series of large humanitarian shelters were erected. At the time of this research, there were thirteen shelters coordinated by the UNHCR across Roraima with a combined capacity of over 9,500 people (UNHCR, 2021). In addition to these, there were several other sites that worked to accommodate Venezuelans. This paper investigates two such sites of humanitarian care that were constructed in response to growing homelessness. The first is a transit centre located at the border and the second is a dormitory in the capital that provided overnight accommodation to those without access to shelter.

By unsettling the banality of (dis)comfort within these two humanitarian spaces of care, I reveal the workings of humanitarianism in everyday contexts of displacement. To begin, I situate this research within the security-humanitarian framework developed by critical humanitarian scholarship. I unpack the meanings and practices of care, comfort, and control, and explore the spatialities and infrastructures that shape migrant mobilities. Following this, I provide a discussion of my research methods and the methodological implications of addressing 'comfort as privilege' from a privileged positionality. I then introduce the two key sites of analysis and highlight the salience of considering a rather mundane element of everyday life: the weather. I move on to discuss differentiated comfort, revealing the ways in which comfort circumscribes the boundaries of care and 'keeps strangers distant' (Pallister-Wilkins, 2018a, p. 997). Following this, I interrogate spatial and temporal practices of containment, highlighting that comfort plays a significant role in the making of mobility governance. In my concluding thoughts, I consider how we might reimagine humanitarian care to normalise comfort (rather than discomfort) in emergency planning.

## 2. Situating Care, Control, and Comfort

Humanitarianism is largely understood as a moral project of compassion. In practice, it seeks to alleviate human suffering and provide care to those in need. Through delivering aid and shelter, humanitarianism appears to remain neutral in both politics and in conflicts (McCormack & Gilbert, 2021, p. 180). Humanitarians themselves are seen to be driven by a personal desire and responsibility to be 'part of something greater than themselves, to help, to be actors in the lively world' (Malkki, 2015, p. 4). When we envision humanitarianism, we conjure up images of certain actors: Médecins Sans Frontières, the United Nations, the International Committee of the Red Cross; we see them performing lifesaving surgeries, delivering essential food and goods that will save lives, and building shelters that will protect 'the vulnerable' from life threatening environments. Humanitarianism saves lives, rescues, protects; it devotes, and it *cares*.

Yet humanitarianism is not quintessentially *caring*. Rather, it manifests alongside the separate but interlocking logic and practice of control. Scholars have critiqued the paradoxical politics of humanitarianism, articulating this underlying tension (Fassin, 2012; Sheller, 2012). They show how, in many ways, exclusionary policies are often justified through humanitarian logics that reproduce 'inequalities and racial, gendered, and geopolitical hierarchies' (Ticktin, 2011, p. 5); what Miriam Ticktin describes as an *antipolitics* of care.

These inequalities and hierarchies typically emerge from humanitarianism's relationship with state powers. States may activate humanitarianism – or vice versa – to respond to emergency. However, the logic of caring for life becomes entangled with aims to control it. As both Simon Reid-Henry (2014) and Polly Pallister-Wilkins (2018) put forward, care and control are in fact constitutive of the modern state, which aims to govern in the most efficient way possible. In so doing, states working in cooperation with humanitarian agents seek to care and protect 'vulnerable' populations, such as refugees or people seeking asylum, while simultaneously securing and protecting national political order (Pallister Wilkins, 2018, p. 994). This relationship between care and control materialises along the security-humanitarian nexus, whereby people on the move are not only framed as 'vulnerable' or 'at risk', but also as potential threats or 'risky subjects' (Aradau, 2004). In these contexts, states and humanitarian actors seek to govern and control the lives of the populations they aim to protect by managing their 'risky' mobilities.

Several studies have investigated the strategies and tools used to govern migrant mobilities at borders (Mountz in Johnson et al., 2011; Stierl, 2017), through borderwork and bordering practices (Parker & Vaughan-Williams, 2012; Rumford, 2012), and across borderscapes (Brambilla, 2015; Pallister-Wilkins, 2018b; Perera, 2007). From these analyses, we learn that borders are not simply confined to territorial markers but materialise across strategic spaces to (often violently and repressively) control, securitize, and capture risky mobilities. Some scholars have tackled the spatialities of mobility governance by exploring refugee camps or informal spaces of shelter, care, and control (Agamben, 2005; Martin et al., 2020). Others have showed how humanitarian spaces have similarly become semi or quasi-carceral spaces, whereby walls, fences, barriers, and military surveillance work to control mobilities (Pallister-Wilkins, 2015). Rather than completely barring mobility, however, these spaces or 'infrastructures' operate to *contain* 'autonomous movement' (Esposito et al., 2020; Garelli & Tazzioli, 2018). Differing from total confinement, containment does not prevent movement entirely, but instead it interrupts, channels, and conditions mobility.

This is where we can begin to interrogate comfort, which plays a fundamental role in humanitarian spaces and spaces of containment. Comfort has emerged as an approach and object of analysis that considers the relation between place, materiality, emotion, and sense (Price et al., 2020, p. 2). Scholars have investigated what it means to *feel* comfortable (Agbedahin & Akalu, 2021), exploring embodied experiences of identity and belonging (Ahmed, 2007; Yarker, 2019). While others have attuned to the architectures of comfort that reveal 'micro-geographies' of care or exclusion, such as park benches and 'anti-homeless spikes' (Petty, 2016; Rishbeth & Rogaly, 2018). It is at this point of departure that I investigate the spatialities of comfort and how it operates, rather than how it is sensed emotionally.

Despite growing scholarship on comfort, it has mostly been 'overlooked as unimportant, trivial and mundane' (Price et al., 2020, p. 18). Indeed, everyday experiences of comfort are often taken for granted, or merely relegated to imaginaries of luxury and opulence (Ferreira, 2021). While perhaps related, luxury is best understood as 'great comfort' or 'abundance', implying a step-up from comfort itself that involves inessential wealth and expense (Cambridge Dictionary, 2024; Merriam-Webster, 2024). By misunderstanding comfort as synonymous with luxury, the implications of (dis)comfort are neglected or perceived as inconsequential. As Zara Ferreira (2021, p. 4) suggests, however, comfort's shift from elitism to egalitarianism emerged after the Cold War when governments sought to improve housing shortages and conditions. It was about ensuring basic needs were met at the 'first stage of comfort': sanitation, plumbing, and heating (Ferreira, 2021, p. 5). These basic needs are similarly incorporated into human rights frameworks that focus on what it means to have 'adequate housing'. The Universal Declaration of Human Rights (1948) maps the requirements for adequate housing, including security, availability of services, accessibility, location, cultural adequacy, affordability and habitability. Importantly, it describes habitability:

> Housing is not adequate if it does not guarantee physical safety or provide adequate space, as well as protection against the cold, damp, heat, rain, wind, other threats to health and structural hazards.

Building on this, the UN Committee on Economic Social and Cultural Rights clarified the right to adequate housing in General Comment No. 4 (1991), expanding the above definition. The approach driving the framework again focuses on 'adequacy', for example, 'adequate space […] adequate basic infrastructure'. It states:

...the right to housing should not be interpreted in a narrow or restrictive sense which  equates it with, for example, the shelter provided by merely having a roof over one's  head [...] Rather it should be seen as the right to live somewhere in security, peace and       dignity.

What enables this right to live and what determines adequacy can be identified through the aspects listed above, including security and habitability. I would also like to highlight another crucial element of the right to adequate housing in point 8(b): the availability of services, materials, facilities and infrastructure. Importantly, comfort is explicitly named here: 'An adequate house must contain certain facilities essential for health, security, comfort and nutrition' (Office of the High Commissioner for Human Rights, 1991, p. 2). What follows is a list of some examples, including, heating and lighting, sanitation, and safe drinking water. While comfort itself is not clearly defined within these frameworks, it materialises alongside key aspects related to housing: habitability, physical safety, protection from weather and threats to health and structural hazards, having enough space and access to sanitation, lighting, and heating/cooling. My understanding of comfort aligns with these spatial and environmental concerns. Comfort requires adequate safety, protection from thermal extremes, and spaces of privacy that do not cause health issues or have structural risks. Like adequate housing, it requires more than providing a roof or basic shelter, but enables security, peace, and dignity.

The characteristics of the right to adequate housing extend beyond formal housing and similarly apply to emergency and displacement accommodation or shelter. The UNHCR Global Strategy for Settlement and Shelter (2014) states that:

> A refugee shelter is first and foremost a home. More technically, UNHCR defines a shelter as a habitable covered living space providing a secure and healthy living environment with privacy and dignity. Refugees have the right to adequate shelter in order to benefit from protection from the elements, space to live and store belongings as well as privacy, comfort and emotional support.

Again, we see that comfort appears to be an important in emergency contexts. The UNHCR standard (2024) is to provide shelter that protects from the weather and offers 'privacy, dignity, comfort, and emotional security'. The Sphere Handbook (2018) also provides the minimum standards in humanitarian responses. It similarly draws on the right to adequate housing but provides practical information as to what this might look like in emergency contexts. One of the shelter and settlement standards, for example, is to provide: 'a basic roof and walls for occupants and their household assets, offering physical security, dignity, privacy and protection from weather [...] optimal lighting conditions, ventilation and thermal comfort' (p. 254). Importantly, however, while comfort is referred to within the handbook, it explicitly refers to *thermal* comfort, noting that 'space heaters and coolers will create suitable living conditions' (p. 260).

Despite these mentions of and aspects related to comfort, what it means and how it functions is typically neglected from research on refugee shelter or spaces of humanitarian care. Like the Sphere Handbook, scholarship that does focus on comfort in humanitarian shelter is often limited to thermal comfort (Lian, 2024; Ullal et al., 2022). It does not consider the ways in which comfort might materialise in other spatial designs, for example, tent layout, flooring, and privacy.

In 1992, Brazil ratified the Covenant on Economic Social and Cultural Rights, which means that the elements of adequate housing are all legally binding for anyone residing in the country, including migrants. This, however, is not binding for the design of

humanitarian shelter, despite the standards set by the Sphere Handbook and UNHCR. Unlike 'adequate housing', many shelters are designed to be temporary and temporarily accessed. The Manual for the Design of Temporary Collective Accommodation in the Americas (UNHCR/ACNUR, 2023) provides guidelines for facilities that shelter people for a limited period of time in emergency contexts, such as transit centres. The main design criteria for accommodation are personal safety associated with violence but also infrastructural risks, disability accessibility, adequate lighting and ventilation, and access to basic water and sanitation (p. 43). For collective spaces, it is recommended that partitions are used every 4-6 people to ensure privacy (p. 55). Interestingly, the term comfort is used several times throughout the manual, but never with regards to accommodation spaces. It is recommended that psychosocial support areas and coffee shops/rest areas for employees are comfortable and have natural lighting and ventilation; comfortable furniture is placed at breast-feeding stations; and adequate thermal comfort is provided in common areas.

It is imperative that we begin to investigate comfort given that there are clear humanitarian standards that explicitly and implicitly describe the design features of comfort in temporary spaces, but there is little research about how this looks on the ground. Thinking through comfort as the spatial and environmental characteristics that condition the habitability, security, privacy, and dignity of shelter, we can understand comfort as a quotidian aspect fundamental to human rights. By adopting this lens of comfort to study humanitarian spaces of care, we can interrogate inherent power relations that shape who has access to comfort and who does not (Price et al., 2020, p. 9). As Laura Price and colleagues (2020, p. 10) explain 'the design, consumption and regulation of comfort: the heating and cooling of air, water and materials is deeply entwined with global systems of spatial inequalities'. By examining the politics of humanitarianism, manifestations of the humanitarian-security nexus are revealed, whereby comfort plays a significant role in the spatialities of care and control.

## 3. Methodology

I conducted this research in the northern Brazilian state of Roraima, where I spent two months between October-December 2019. I stayed in the capital of Boa Vista and volunteered with a local humanitarian organisation that responds to the needs of those living without adequate shelter. My aim was to further understand the dynamics of military-humanitarian governance and I therefore focused on the institutions, actors, and spaces that shaped the delivery of aid that consequently had implications on migrant mobility.

A limitation of this work was being unable to develop ongoing relationships with people who were living without adequate shelter and accessing aid from this organisation. This was due to several reasons, including the format of my work in the office, whereby I focused on translating documents, entering data, and writing success stories. As with institutional ethnography (Billo & Mountz, 2016), I conducted participant observation by 'following' volunteers and staff during their daily tasks, including visiting humanitarian sites of care and attending stakeholder meetings. On several occasions, I also assisted with handing out hygiene kits at the office, although these were only given as once-off resource, and I soon reckoned with the challenge of building and establishing trust in this format. Recognising the methodological implication of doing no harm (Mackenzie et al., 2007), I turned to the spatial governance of the humanitarian response. I paid particular attention to the spatialities of shelters, drawing on Annette Miae Kim's (2015) spatial ethnography to consider the built

environment in relation to site management and social dynamics. I hand-drew maps and wrote extensive field notes, triangulating this with data from interviews and informal conversations with thirty-five humanitarian staff, volunteers, military personnel, taxi drivers, and Venezuelan migrants.

I first started thinking about (dis)comfort when I moved through humanitarian shelters. I observed what I imagined was uncomfortable: hard concrete floors and hundreds of cramped tents. I also felt discomfort, with sweat forming at my brow just minutes after stepping outside a cool office. And I talked about comfort, often implicitly, for example with an aid worker who noted people 'roaming' the streets in search of shade. Comfort, therefore, emerged in correspondence with others, as an embodied experience, and as a visual observation. This prompted me to consider what it means to study comfort.

Comfort, as we have seen, is a negotiated process that unfolds within and in relation to social worlds (Price et al., 2020, p. 6). It is driven by imaginaries of what is good, safe, and easy. As Holliday (1999, p. 489) attests, comfort is 'an easy, unthinking state [...] being comfortable [...] implies a lack of necessity to worry about the world or one's position in it.' From the outset, comfort is inherently privileged. To live a life of comfort is reserved only for some. My own positionality as a white, cis woman with funds to complete a PhD in Australia reveals that I hold this privilege. Indeed, I benefit from the systems that enable me to consider the implications of comfort, rather than my own basic survival. Like Price and colleagues (2020), I worried that focusing on comfort might only uphold such privileged narratives. However, by reckoning with the socio-political operations of comfort, I hope that we might be able to 'imagine worlds otherwise' (Price et al., 2020, p. 17).

While comfort is understood as a privilege, its absence is justified through notions of luxury. Striking levels of discomfort have become normalised across the globe, particularly within contexts of forced mobility. Despite the Universal Declaration of Human Rights (1948), basic elements of environmental and physical comfort are often considered secondary to providing emergency solutions. I challenge notions that comfort comes after necessity and maintain that at its most basic level, comfort is a human right. In the context of this research, humanitarian spaces and emergency accommodation reveal the implications of neglecting comfort in the design of spatial infrastructure. As this paper will demonstrate, spatial (dis)comfort has the ability to reinforce social divisions, to keep people separate, and to uphold narratives of deservingness. By considering comfort in the care-control nexus, we can recognise its role in the governance of risky mobilities and how it is weaponized to contain and securitize. As such, discomfort must not be normalised. To investigate and potentially enhance emergency responses to migration, comfort must be taken seriously at the most basic level.

## 4.   Revealing the mundane in everyday humanitarian spaces

Pacaraima and Boa Vista are some of the few Brazilian cities located just north of the equator. The weather is typically hot and humid year-round, with months of heavy rains and flash flooding. Taking the weather into account during emergency planning is crucial, and research continues to highlight extreme thermal discomfort in shelters and camps (Albadra et al., 2017; Domínguez-Amarillo et al., 2021). As I walked through a UNHCR shelter in Boa Vista, an aid worker explained that inside temperatures of the refugee housing units made of steel and metal could reach up to 45 degrees Celsius during the day (113F). This meant that most people were only able sit in the shaded area

near the offices, where the army – providing security – would surveil and patrol on foot. This brief example illustrates the significance of studying the implications of (dis)comfort in the everyday interaction of humanitarianism, space, and the environment.

The first site of analysis in this paper is the BV-8 transit centre located in the border city of Pacaraima. It is managed by the Operation Welcome task force in partnership with UNHCR and several humanitarian agencies. The army is present on site to provide security. It provides temporary accommodation to Venezuelans as they cross the border into Brazil while they make other living arrangements or travel plans. The centre accommodates up to one thousand people at a time. Individuals are permitted to stay for a maximum fifteen days, which ensures enough time for second dosages of vaccines to those requiring them. Importantly, BV-8 was constructed in response to growing homelessness amongst migrants and refugees in Pacaraima, given that the UNHCR shelters in the city were typically at capacity. Operationally, it serves as a humanitarian space of care. For an aerial view of the site, see figure 1 below.

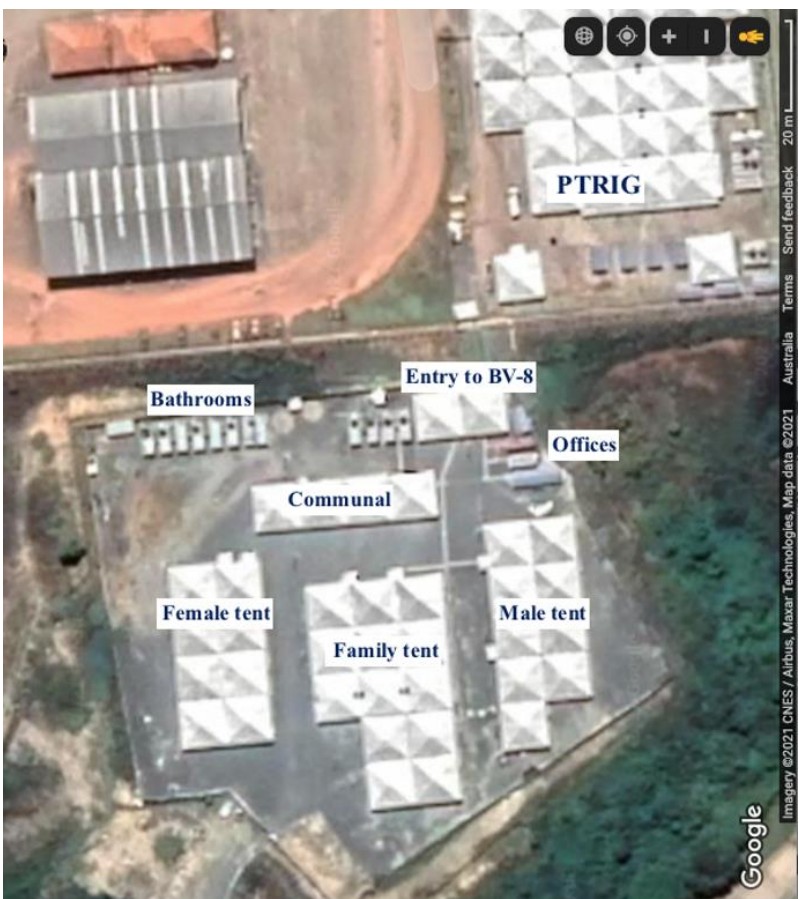

*Figure 1: aerial view of BV-8 centre, Google Maps 2019*

In BV-8, there are three large tents that each accommodate over 300 people. They are divided between men, women, and families. Single mattresses are available for use on the wooden floors, or in some cases, on metal bunk beds. Clearly marked Operation Welcome signs are posted throughout BV-8, signalling each area's designed purpose. This includes a covered communal space in the centre, where meals are served and people can gather. There are three meals provided per day, and there is access to bathrooms, laundry sinks, and medical care (Presidência da República: Casa Civil n.d.).

The second site is the 'rodoviária', which functioned as a type of homeless shelter in the capital of Boa Vista. However, the word 'shelter' was rarely used to describe this site. Instead, it was often referred to as a 'support post', 'overnight area', or 'refuge' (Boa Vista Já, 2019; Correia Lopes, 2019). Given that it operated on an overnight, first-come, first-serve basis, I will refer to it as an overnight dormitory. As humanitarian responses are always dynamic, the dormitory was replaced in 2022 with a site just a few hundred metres away. The new space, a Reception and Support Post (PRA), still operates on an overnight basis, but with some key differences. The rodoviária dormitory was overseen by the local municipality with help from Operation Welcome, while the new PRA is run by IOM in collaboration with Operation Welcome. At both the old and new sites, the army provided and continues to provide security, monitoring, and surveillance. Despite the construction of the new site, the rodoviária dormitory remains a significant point of analysis, given that it operated for almost four years and provides insight into the humanitarian spaces of care.

The rodoviária dormitory was closed during the day and could accommodate up to one thousand people. Within the site, there were rows of tents at various sizes. The single-person tents made up the majority of space, but I was told they typically slept two people or a couple with a child. The family-sized tents were slightly larger and designed to accommodate six-eight people or two separate families. There were also four large tents allocated for categorically vulnerable individuals, including elderly persons, pregnant or breastfeeding women, and disabled people. The site was located within a fenced concrete area behind the international bus station and included several portable toilets. See figure 2 for a hand-drawn map of the site. While the new PRA site offers some welcome changes, including metal bed frames with mattresses, these are all lined up under a large, shared marquee and no privacy curtains are provided. Though further spatial research is needed, this configuration suggests there are still questions about designing comfort.

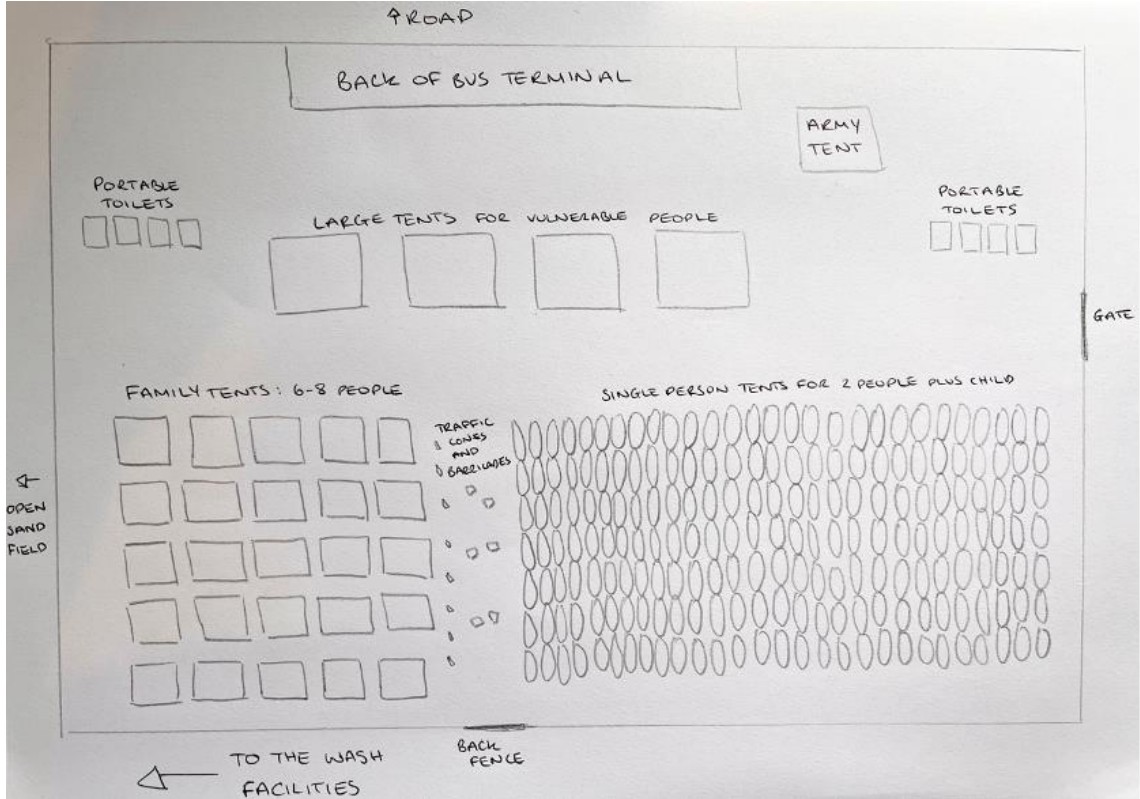

*Figure 2: layout of Rodoviária dormitory, drawn by author January 2020*

## 5. Differentiated Comfort

*We approach the metal barricades of the entrance and get checked in by the two army officers on duty. The heat feels overwhelming and as we walk, the small grey stones beneath our shoes crunch and shift unevenly with every step [...] Moving past the empty communal area in the centre of the site, I see people sitting in the shadows created by the large tents, likely seeking some minimal reprieve from the unforgiving temperature [...] When we finish the tour, we drop into one of the humanitarian agencies offices, situated near the entrance. As the door opens, a rush of cold air escapes and we step into the comfort of the air-conditioned room.* (Author reflections from fieldnotes January 2020).

In humanitarian spaces, hierarchies of comfort are revealed. In the excerpt above, I reflected on my visit to the BV-8 transit centre where I was given a tour by an aid worker based at the site. While the warm weather was not unusual for this region, I immediately noticed the differentiated levels of comfort that were available. After stepping into one of the large tents, it only seemed to hold in the heat. It made sense that just a handful of people remained inside while the rest sat out in the shade. This scene was strikingly distinct from the small humanitarian office only a few feet away, where our team chatted with other aid workers in the cool air after the tour. A similar image emerged when I visited the rodoviária overnight dormitory:

*By the time we leave, the mid-morning sun is beating down on us. The heat and humidity constantly remind me how close we are to the equator [...] The dust of the dry season forms a layer of dirt over the road, and the unevenness of the ground makes me wonder how the puddles must form in the wet season... [The UNFPA staff member] leads us into a small army tent and I welcome the coolness of the air conditioner. There is one long table with benches, a fridge, and a gas cooker.* (Author reflections from fieldnotes January 2020).

After visiting several humanitarian spaces in Boa Vista, I noticed a pattern emerge: there were often very limited cooling options and protective measures from the natural elements in spaces designed for people seeking safety. This was rarely the case for spaces intended for military and humanitarian agents. Rather than being completely devoid of discomfort, these military/humanitarian spaces offered infrastructure that would provide *more* comfort than others. While the broad categories of 'humanitarian worker' and 'migrant' or 'refugee' do not necessarily capture the nuances of identity, these spaces illustrate differences in the degree to which comfort is available. Comfort can thus be understood as a scale (Agbedahin & Akalu, 2021, p. 221); and its levels can reveal fundamental truths about humanitarianism.

*...in Brazil now we don't consider this as a camp because they are not supposed to stay there. It's a temporary shelter [...] we have this shelter, we don't call it a shelter because there is no structure as a shelter. We call it like a dorm, temporary dorm, that is BV-8.* (Interview with a UNHCR staff member November 2019).

Humanitarianism, in essence, is a temporary practice. The UNHCR staff member above described this temporariness, first by noting how the 'shelters' are only meant to be temporary, and then by explaining how the BV-8 transit centre could not be conceived as a shelter, given that it operates as a temporary dormitory. Despite efforts of development and integration or evidence of long-term, protracted situations, humanitarian intervention is designed to be temporary. Tents are often the first to be used in emergency responses, given their compactness, weight, and cost. Tents are also undeniably temporary; just as easily as they can be assembled, they can be quickly

dismantled. This is often crucial for governments demanding temporary solutions and for local communities similarly opposed to suggestions of permanence evoked by more solid structures (Scott-Smith, 2024, pp. 30−35). Depending on the context, providing shelter may exceed local living conditions and lead to tensions (Sphere Association, 2018, p. 8). This could be said of Brazil, where the boost in funds at the border marked a visible division of resources (Moulin Aguiar & Magalhães, 2020). These were not necessarily limited to accommodation but were somewhat representative of the government's sudden willingness to provide support to incoming migrants, while continuing to neglect a wider impoverished region.

Despite how these structures might appear, we have already seen some of the concerns within these spaces of humanitarian care. This includes, for example, limited protection from the heat, lack of privacy, hard concrete flooring, and overcrowding. These elements, which can be understood as elements of discomfort, have become normalised and expected in displacement contexts. This is a result of the underlying functions of humanitarianism: to save lives and alleviate human suffering (Fassin, 2012). Firstly, to save lives requires efficiency; practices and procedures that order populations are essential. These are inevitably entangled with the calculations and management that come with mobility governance – only then can systems of saving lives be efficiently delivered (Pallister-Wilkins 2018a, p. 1001). Equally, shelter must be designed and constructed as efficiently as possible. Not only constrained by the politics of temporariness, shelters must be affordable, light weight, easy to deliver and construct. Spaces of shelter must also be designed in a way that enables supervision, making it easier to manage the population (Martin et al., 2020). Shelter must therefore be designed, constructed, and managed efficiently. Efficiency saves lives. Comfort, while fundamental to living, does not. This explains, in part, why comfort is referenced in humanitarian standards, but not delivered in practice.

Secondly, given that humanitarianism strives to alleviate human suffering, suffering cannot be separated from it. In order to be 'eligible' to receive humanitarian aid, one must experience suffering. This is exemplified in the 'success stories' I was tasked with writing in Brazil, which were sent to international donors. These were aimed at first highlighting the suffering of 'vulnerable' people, followed by the ways in which delivering aid had alleviated this suffering. Indeed, suffering is so fundamental to humanitarianism that it is extracted and exhibited as a means of securing the future of aid work – often seen in advocacy campaigns (Cabot, 2016). As such, suffering largely defines humanitarianism; it is both normalised and expected. This carries through the times and spaces of intervention, whereby refugee shelter and sites of care are designed to alleviate suffering, to some extent, but to not go any further so as to provide comfort. As Tom Scott-Smith (2024, p. 37) demonstrates, this is because those on the receiving-end of aid are constructed as a 'different category of person', where they 'should not expect to live in such high-quality accommodation'. For Venezuelans in Brazil, this is exacerbated by assumptions of comfort-as-luxury that is reserved for a specific socio-economic status, not for those from low and middle classes arriving at the border.

By thinking through comfort in humanitarian practice, we can see that it functions to circumscribe the boundaries of care; differentiating who has access to comfort and who does not. A division is thus created through humanitarianism. Didier Fassin (2007) examines this division by showing how humanitarian intervention gives value and meaning to human life; it is a politics of life. Though he describes several elements of this, here, I focus on just one. By siding with 'victims' and defending their cause, humanitarian agencies produce a distinction between:

those who are subjects (the witnesses who testify to the misfortunes of the world) and those who can exist only as objects (the unfortunate whose suffering is testified to in front of the world) (Fassin, 2007, p. 517).

This distinction between the value of human life can be located in the differentiated spatialities of comfort in humanitarian spaces. To illustrate this further, I briefly turn to another site located at the border: the Brazilian army base.

The army base is located near the BV-8 transit centre and the two sites are separated by the triage centre, as pictured in figure 4. Military, government, and humanitarian personnel may access the army base for accommodation, offices, and storage supply. Despite their close proximity and similarities in design (including large tents, uniform layout, fenced perimeter), the spatialities of these two sites are starkly contrasted with one another. When I arrived at the army base, I noticed a man in uniform mowing the green, well-maintained lawn. The large tents I visited were mostly for storing supplies from humanitarian agencies. For accommodation, there were neat rows of small dwellings constructed from repurposed shipping containers. I walked through the rows, observing colourful pot plants placed out the front doors and air-conditioning units fitted to the sides. As I noted in my field diary, the army base felt 'semi-suburban'. Unlike BV-8, there was a sense of comfort in the design, rather than sterility.

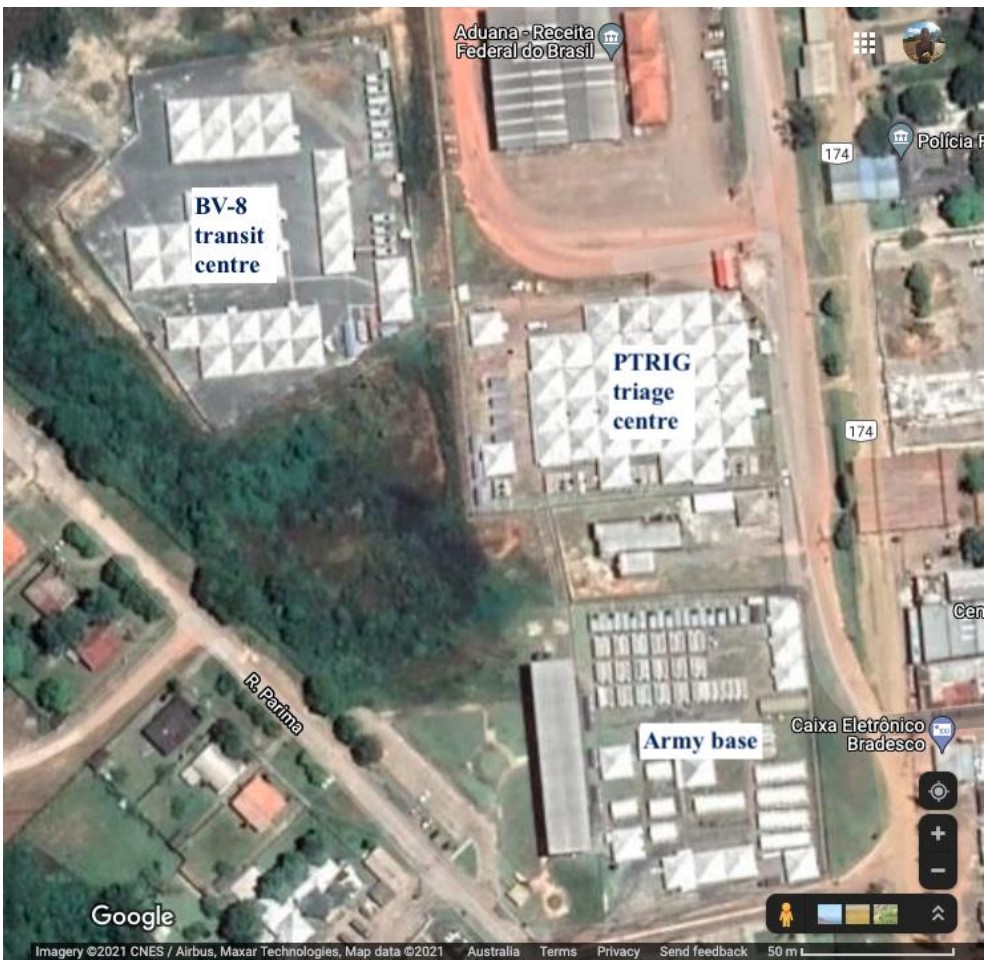

*Figure 3: aerial view of the army base, triage centre, and BV-8, Google Maps 2019*

I mentioned this in conversation with a UNHCR leader for security. They explained that it was common practice to create spaces of comfort in army bases around the world, noting having seen soldiers planting rose bushes in Pakistan. However, this same comfort – particularly protection against natural elements that provides privacy and dignity (UNHCR, 2024) – was not discernible in the design of the BV-8 centre. Instead, the spatialities evoked a sense of anonymity commonly associated with refugee camps designed to order and supervise populations (Martin et al., 2020, p. 754). The communal area in BV-8, for example, is located in the centre of the site, which makes it ideal for security personnel to surveil and monitor the population. There are tables and chairs under tall marquees that offer some immediate reprieve from the heat. However, the surrounding walkways are not covered and there are no fans provided on site. Considering the BV-8 centre and the army base together highlights how only basic care is available to Venezuelans seeking safety, while comfort is limited to the army base and humanitarian offices.

The differences in spatial design reveal how comfort circumscribes the boundaries of humanitarian care by 'keeping strangers distant' (Pallister-Wilkins, 2018a, p. 997). While historically, colonial humanitarianism meant governing from afar (Laidlaw, 2012), it is now common to save lives up-close, for example, by providing healthcare or building shelters (Pallister-Wilkins, 2018a). There was once an underlying expectation that humanitarians should be 'close' to suffering, to experience it. This idea may still linger, at least geographically, however, the professionalisation of humanitarian work has created and maintained social distance. This distance comes as humanitarianism works to respond efficiently (Reid-Henry, 2014; Pallister Wilkins, 2018a). Distance is created and maintained between the humanitarian actor and the humanitarian subject. The actor is perceived as useful, virtuous, passionate, and capable of saving lives, while the subject is constructed as an unequal 'other' and a 'victim with needs rather than a person with full subjecthood' (Pallister-Wilkins, 2018a, pp. 997–998). By limiting comfort to spaces for military and humanitarian actors, the distance between them and people seeking safety is reinforced. 'Distant others' are provided access to basic needs and basic care, while military-humanitarian agents are afforded additional access to comfort.

Further to highlighting the divisions between those delivering and receiving care, investigating (dis)comfort reveals its everyday implications. For people seeking safety, navigating spaces that provide protection from the heat and monsoon rains is a daily task. This extends beyond any time spent at the transit centre or the overnight dormitory, particularly for those without access to secure housing. As noted in my field diary, I often observed men, women, and children sitting or lying under trees by busy roads or inside small rooms where ATMs were located. This illustrates how discomfort permeates the quotidian experiences of Venezuelans who do not have access to rent or relocate. This includes those living on the streets, in spontaneous occupations, as well as those staying within formal shelters. Comfort, then, is an undeniable characteristic of humanitarian intervention.

## 6. The (dis)comfort of containment

*Here in Brazil, [there is] a difference between other operations because we really do have the support from the government [...] If you compare this operation for other operations, even in Europe we don't have this kind of support from the government, especially from the army. Normally the army is not the one helping, [but] is the one mostly just [providing] security for the borders [...] but we have really good support from the army. The army in Roraima specifically is focused in [engineering]. So they*

*help a lot. They actually are the ones doing all the [engineering] and structural things inside of the shelters.* (Interview with UNHCR staff December 2019).

Brazil's military plays a key role in the Operation Welcome task force. In addition to securing the border and other urban spaces, they also design and construct humanitarian sites. As expressed by the UNHCR staff member, this support from the government and military has been crucial for the response, particularly for providing spaces of care to those seeking safety. However, as Weizman (2012, p. 4) attests, 'every political act is registered in space'. Thus, the architecture of humanitarian spaces, including their design and construction by military forces, reveals underlying logics of securitisation that aim to control 'risky' mobilities. In military-humanitarian spaces, Weizman (2012, p. 4) suggests, 'spatial organizations and physical instruments, technical standards, procedures and systems of monitoring […] have become the means for […] governing the displaced, the enemy and the unwanted'. Examining the BV-8 centre and the rodoviária dormitory reveals that while these spatialities are not solely founded on logics of securitisation – given the simultaneous aims of providing care to vulnerable populations – they are almost always co-opted into securitisation projects.

I have elsewhere discussed how the spatial practices of the humanitarian response reveal a logic of mobility governance, that is: debilitating mobilities (Alexander, 2023). This logic captures the interplay between biopolitical and necropolitical mobility governance that is neither designed to fully prohibit mobility, nor freely allow it. Instead, autonomous mobilities are contained – interrupted, conditioned, and controlled – as a means of regulating risky populations (Esposito et al., 2020; Garelli & Tazzioli, 2018). As one Brazilian humanitarian staff member explained: 'the shelters are made for the migrants to stay just for a while, that also means the structure and the rules are very coercive. The shelters do not help to make some autonomy and [empowerment] for these families' (October 2021). This containment is justified and disguised by the basic humanitarian care that is simultaneously provided, rendering the containment a mechanism of debilitating mobilities. While this is key for understanding the spatialities of the dormitory, I would like to further draw attention to the role of comfort in this containment.

We have so far seen that comfort plays a role in care by circumscribing its boundaries. Here, we see that (dis)comfort is fundamental to containment, which is a key method of control. At the rodoviária dormitory, the tents were positioned so tightly next to one another that moving between them proved difficult. When I visited, it was during the day while the dormitory was closed. I walked alongside two Venezuelan migrants, one volunteer and one staff member working for different aid agencies. They suggested that it was better that I visited when the dormitory was closed, as once it opened it would quickly become very loud and overcrowded, and aid workers were sometimes met with verbal harassment.

Quiet now, and faced with a sea of camouflage tents, I was forced to move around the perimeter, so as not to trip on the edges or disrupt their configuration. The tents were erected on bare concrete and dirt grounds; no mattresses were provided, save for the few I noted in the tents allocated for categorically vulnerable persons. The tents also accommodated more people than intended, with families sharing with other groups of people unknown to them. These spatialities were quite clearly uncomfortable and cramped. Furthermore, there were no fans, and the marquees only provided shelter directly above the tents and as such, were ineffective against the rain when walking to the portable toilets or spending time outside prior to the sleep curfew.

The design of this site worked to contain the population sleeping there. Much like institutional camps, the rodoviária dormitory deployed strategies to control and surveil a 'risky' population (Martin et al., 2020, p. 753). These were people on the move, already representing a general risk to public security (Hyndman & Giles, 2011), but they were also people whose mobilities posed a greater risk since they were not confined to formal refugee shelters. Having inadequate access to shelter, or experiencing homelessness, meant that their mobilities were more difficult to pin down and manage (Alexander, 2024, p. 39). The dormitory provided a confined space that could enable the temporary containment of these hyper-mobile and risky mobilities. Its features, the tight layout of the tents, security barriers, fences, entry and exit controls, and regular military patrols, worked to control everyday mobilities (Pallister-Wilkins, 2015, 2018a). There were few ways that the dormitory could be 'differently accommodated and utilised' by the people coming to stay overnight, making it very difficult to resist its overarching governance (Martin et al., 2020, p. 755).

The spatialities of containment are further entangled with temporalities. The dormitory would open at 4pm on a first-come, first-serve basis. After waiting outside, individuals were processed at an initial checkpoint. Those who met the requirements of refugee or temporary residency status could enter the dormitory provided there were vacancies. After storing belongings in lockers, individuals were expected to sleep in cramped conditions at the 10pm curfew. The following morning, all people (except for categorically vulnerable persons) were required to exit by 7am-8am. These routines were imposed by the local municipality through the military personnel on site. They enforced a structured rhythm of everyday life that enabled easier management and closer control over 'risky' mobilities (Coletta et al., 2020; Turnbull, 2014). By disciplining time in spaces of discomfort, governing agencies limited the ways in which migrants could enact political agency (Turnbull, 2014). By doing so, they sought to maintain control over the movement of people in, out, and through humanitarian spaces (Garelli & Tazzioli, 2018).

What these spatialities and temporalities reveal is that comfort plays a significant role in the practices of containment. It is, however, the *absence* of comfort that makes this containment so insidious. It is not simply that mobilities are disrupted and directed in place, but that they occur within an environment that limits and restricts comfort through a lack of privacy, structural hazards, and protection from natural elements. These features, so entangled in containment, reveal the operation of debilitating mobilities, whereby spaces such as the rodoviária dormitory are deemed humanitarian sites of care, but that fundamentally function to control mobile populations. Comfort, or discomfort rather, is layered into this approach of mobility governance; one cannot think of spatial and temporal mechanisms of containment without considering the role of comfort. While it is yet to be said whether populations can be contained comfortably, taking comfort as a lens provides insight into logics of control, and within that, we see traces of deterrence.

Providing only basic amenities while restricting comfort reflects a wider logic of the humanitarian response, that is, to deter people from becoming accustomed to ongoing humanitarian care. This is more explicitly illustrated with an example from another refugee shelter in Boa Vista, where I spoke with a Brazilian aid worker based there. They described how humanitarian staff had recently installed sails above the site with the aim of decreasing hot temperatures and heavy rainfall. Following this, military personnel removed the sails with the claim that residing individuals would become 'too comfortable'. This would result in them relying on humanitarian shelter and staying

long-term; though this did not seem to concern the aid workers themselves, who attempted to provide extra protection but were unable to overcome the limits set by the Operation Welcome taskforce. Similar logics of deterrence that weaponize comfort emerge elsewhere. At the US-Mexico border, freezing cold holding cells are used to deter individuals from seeking asylum (Riva, 2017, p. 317). Those forced to cross the Sonoran Desert of Arizona are faced with a 'hybrid collectif' of deterrence that includes harsh temperatures and terrain (de León, 2015, p. 60). While more deleterious than removing sails from the refugee shelter, a similar logic of deterring Venezuelans from becoming too comfortable emerges.

The logic of deterrence at the overnight dormitory was not necessarily about discouraging long-term settlement but about ensuring people did not become 'too comfortable'. By this logic, creating spaces of comfort could lead to long-term use of the site and reliance on humanitarian aid, rather than moving to rental accommodation or even to another state outside of Roraima. Indeed, when I sat in on a meeting with key stakeholders in Operation Welcome, an army colonel affirmed that their main priority was the relocation of Venezuelans to other parts of Brazil. While this program is voluntary, in that people can choose if they want to move, they have no control over where, when, and how (Medina Araújo & Ramos Barros, 2025, p. 104). This highlights the underlying logics of mobility governance that permeate the entire response and that consequently lead to the design of spaces without basic elements of comfort. It further reveals how control and security are anchored in military-humanitarian intervention. Regardless of whether comfort is present or there is an absence of comfort, it plays a significant role in which it may be weaponized in the practices of controlling and containing migrant mobilities.

## 7. Conclusion

This article has contributed to the discussion on humanitarian infrastructure as a mechanism of governance and the operation of both care and control. Humanitarian spaces of temporary care are typically designed for basic functionality or adequacy. Despite being embedded within legal and humanitarian frameworks of shelter, comfort is most often neglected in spatial planning. However, bringing comfort into discussion with humanitarian intervention and mobility governance provides a lens to examine practices of care and control. In Brazil, humanitarian spaces are shaped by military intervention at the design, construction, and management levels. Spaces allocated to humanitarian actors are typically offered more access to comfort – by way of cooling systems and protective measures – while sites for Venezuelan migrants and refugees are not afforded the same. As such, we see that comfort circumscribes the boundaries of care; comfort is differentiated between those delivering aid and the 'distant others' receiving it. However, a key limitation of this work was the inability to include the perspectives of people who were living without shelter and were accessing these spaces of care. While I have focused on the spatial and political condition of comfort, future research would benefit from investigating affective and emotional experiences within these spaces.

Comfort, or its absence, plays a significant role in humanitarian spaces that contain, control, and securitize everyday 'risky' mobilities. This has implications for future research beyond the context analyzed here, whereby we may examine diverse state

reception systems and the production of (dis)comfort in spaces where migrants are 'welcomed'. Spatialities that subject migrants and refugees to such discomfort may prompt scholars, designers, and practitioners to think more closely about how comfort may be weaponized as a mechanism of mobility governance. This could be useful in rethinking more inclusive strategies for management and planning.

Given the financial constraints and political entanglements of humanitarian borderwork, is it possible to design and construct spaces that do not reinforce unequal power divisions? Co-designing spaces with people seeking safety may be one way forward. By doing so, spaces of care can be reimagined to both provide protection from harsh climates and cater to socio-cultural needs (Aburamadan, 2022). Importantly, however, this must be considered in the early stages of the design process (Aburamadan, 2022, p. 5), rather than as an afterthought or when displacement becomes protracted. While this approach would counter some of the issues raised in this paper, like Pallister-Wilkins and colleagues (2023, p. 297), I recognise that humanitarianism 'can never be an endeavor devoid of power relations in any form.' To simply redesign spaces without seriously engaging the broader implications of humanitarianism would ignore ongoing injustices embedded within the system of humanitarian intervention. Future spatial designs and redesigns would need to tackle underlying logics of securitization and work to normalize comfort as a human right.

## Acknowledgements

To the many humanitarian staff and volunteers I worked alongside during this project, I am grateful for your openness and support, and I recognise the importance of your ongoing work in Brazil.

## List of figures

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
