# Peer review of "Calibrating Comfort in the Care-Control nexus of Humanitarian Borderwork"

_Migration Politics_

## Round 1 · Referee Report · Anonymous (Referee 1) · 2025-2-3

Strengths

1) The author draws on a broad theoretical corpus, integrating perspectives from migration sociology, critical humanitarian studies, and the literature on mobility governance. The work is grounded in key references such as Didier Fassin, Polly Pallister-Wilkins, and Miriam Ticktin, successfully positioning itself within an established debate while introducing a novel element. 2) The article presents an original analysis of the concept of comfort in humanitarian contexts, a topic often overlooked in the literature on migration governance and humanitarian action. The author highlights how (dis)comfort is strategically employed in mobility control and management processes. This approach connects the issue of the right to adequate housing with the security logics that regulate refugees. 3) The research combines participant observation, informal interviews, and spatial analysis of humanitarian camps in Brazil. The use of hand-drawn maps and ethnographic descriptions enriches the analysis, making it more immersive. Additionally, the reflection on the author’s own privileged position, acknowledging that studying comfort is a luxury compared to the fundamental concerns of migrants, demonstrates a commendable methodological awareness. 4)A convincing discussion of the care-control nexus. The article effectively demonstrates that humanitarian infrastructures are not merely instruments of aid but also of containment. The idea that comfort is selectively distributed (reserved for humanitarian workers and military personnel while refugees receive only the bare minimum) is a strong argument that enriches the debate on humanitarian space as a governance mechanism.

Weaknesses

1)Although the article analyzes comfort as a material and political element, its definition remains elusive from an affective/emotional perspective.

Report

The article should be accepted for publication because it makes an original and well-structured contribution to the debate on humanitarianism and mobility governance, addressing a topic often overlooked in academic literature: the role of comfort in the processes of controlling and assisting migrants. The study is not only theoretically significant but also has practical implications for the design of humanitarian infrastructures and reception policies. The analysis of comfort as a governance mechanism could be useful in rethinking more inclusive and dignified strategies for refugee management. Moreover, it may have important implications beyond the contexts analyzed in this article, for instance, in examining governmental reception systems in different national settings and in analyzing the production of (dis)comfort in contexts where pro-migrant volunteers operate.

Recommendation

Publish (meets expectations and criteria for this Journal)

---

## Round 1 · Referee Report · Anonymous (Referee 2) · 2025-2-4

Report

This article presents a well-researched contribution to humanitarian and migration studies. The article is well-situated within critical scholarship and presents a compelling analysis of the often-overlooked role of comfort in humanitarian borderwork. By examining Brazil’s military-humanitarian response to Venezuelan migration, the author introduces comfort as an analytical lens to explore the interplay between care and control in humanitarian spaces. The study’s methodological approach, including participant observation and fieldwork reflections, enriches the analysis and provide empirical insights on inherent power relations that shape who has access to comfort and who does not.

Addressing the following minor issues would further strengthen the paper’s impact and value:

• The article presents a clear-cut opposition between those who provide aid and those who receive it. While it is established that military personnel and humanitarian workers have greater access to comfort compared to ‘beneficiaries’, is there a difference in terms of the narrative surrounding the (non-)provision of comfort to Venezuelan migrants? What are the implication in terms of claims (if any) and practices on the ground (of migrants as well as humanitarian actors)?

• The article primarily relies on ethnographic observations (the maps drawn by the author are particularly useful), but the methodology also references interviews. The paper would perhaps benefit from including quotes from these mentioned interviews. The voice of migrants and humanitarian actors does not currently emerge from the paper.

• The conclusions could be enriched by discussing the research limitations and outlining the paper's contribution to existing literature on humanitarian borderwork and implications of the findings for future studies.

Recommendation

Ask for minor revision

---

## Round 2 · Referee Report · Anonymous (Referee 2) · 2025-2-18

Report

I have reviewed the revised version of the manuscript. The author has satisfactorily addressed the provided comments. I am pleased with the changes and recommend that the article be published.

Recommendation

Publish (meets expectations and criteria for this Journal)

---

## Round 2 · Referee Report · Giacomo Lampredi (Referee 1) · 2025-2-18

Strengths

The article presents an original analysis of the concept of comfort in humanitarian contexts, a topic often overlooked in the literature on migration governance and humanitarian action. The author highlights how (dis)comfort is strategically employed in mobility control and management processes. This approach connects the issue of the right to adequate housing with the security logics that regulate refugees.

Weaknesses

The weaknesses of the previous version, particularly regarding the discussion of comfort, have been well contextualized and justified. The current version no longer has any weaknesses to report from my side.

Report

My opinion is to accept the manuscript as it is in its current version.

Recommendation

Publish (meets expectations and criteria for this Journal)

---

## Round 2 · Author Response

To the reviewers,
Thank you both for your insightful feedback and your points to strengthen this article. Please see the list of changes available. Many thanks and kind regards.

---

## Round 2 · List of Changes

Reviewer 1:
The article presents a clear-cut opposition between those who provide aid and those who receive it. While it is established that military personnel and humanitarian workers have greater access to comfort compared to ‘beneficiaries’, is there a difference in terms of the narrative surrounding the (non-)provision of comfort to Venezuelan migrants? What are the implication in terms of claims (if any) and practices on the ground (of migrants as well as humanitarian actors)?
- I have included a note that demonstrates the differing views of humanitarian workers and the state-military approach, particularly with regards to the installation of sails in the other shelter. On page 14, I write: 'This would result in them relying on humanitarian shelter and staying long-term; though this did not seem to concern the aid workers themselves, who attempted to provide extra protection but were unable to overcome the limits set by the Operation Welcome taskforce'… More research is needed to ground claims of comparable comfort from the perspective of migrants. I have noted this in the limitations (see point 3).

The article primarily relies on ethnographic observations (the maps drawn by the author are particularly useful), but the methodology also references interviews. The paper would perhaps benefit from including quotes from these mentioned interviews. The voice of migrants and humanitarian actors does not currently emerge from the paper.
- I have included a few more direct quotes from interviews as well as notes from conversations. See for example:
- Page 9: …in Brazil now we don't consider this as a camp because they are not supposed to stay there. It's a temporary shelter […] we have this shelter, we don't call it a shelter because there is no structure as a shelter. We call it like a dorm, temporary dorm, that is BV-8. (Interview with a UNHCR staff member November 2019).
- Page 12: As one Brazilian humanitarian staff member explained: ‘the shelters are made for the migrants to stay just for a while, that also means the structure and the rules are very coercive. The shelters do not help to make some autonomy and [empowerment] for these families’ (October 2021).
- Page 13: When I visited, it was during the day while the dormitory was closed. I walked alongside two Venezuelan migrants, one volunteer and one staff member working for different aid agencies. They suggested that it was better that I visited when the dormitory was closed, as once it opened it would quickly become very loud and overcrowded, and aid workers were sometimes met with verbal harassment.

The conclusions could be enriched by discussing the research limitations and outlining the paper's contribution to existing literature on humanitarian borderwork and implications of the findings for future studies.
- I have now revised the conclusion to include discussion on limitations, my contribution, and implications for future work. Please see for example on page 14: This article has contributed to the discussion on humanitarian infrastructure as a mechanism of governance and the operation of both care and control […] However, a key limitation of this work was the inability to include the perspectives of people who were living without shelter and were accessing these spaces of care. While I have focused on the spatial and political condition of comfort, future research would benefit from investigating affective and emotional experiences within these spaces.
Comfort, or its absence, plays a significant role in humanitarian spaces that contain, control, and securitize everyday ‘risky’ mobilities. This has implications for future research beyond the context analyzed here, whereby we may examine diverse state reception systems and the production of (dis)comfort in spaces where migrants are ‘welcomed’. Spatialities that subject migrants and refugees to such discomfort may prompt scholars, designers, and practitioners to think more closely about how comfort may be weaponized as a mechanism of mobility governance. This could be useful in rethinking more inclusive strategies for management and planning.

Reviewer 2:
- While it was not requested, I have added a note in the limitations of this paper that recognise the need to explore affective/emotional understandings of comfort. I believe this needs to be done with the direct input of those making use of these spaces. On page 14 I write: While I have focused on the spatial and political condition of comfort, future research would benefit from investigating affective and emotional experiences within these spaces.

---

## Editorial Decision

accepted_in_target_journal